# Modeling the epidemiological impact of the UNAIDS 2025 targets to end AIDS as a public health threat by 2030

John Stover[1]*, Robert Glaubius[1], Yu Teng[1], Sherrie Kelly[2], Tim Brown[3], Timothy B. Hallett[4], Paul Revill[5], Till Bärnighausen[6], Andrew N. Phillips[7], Christopher Fontaine[8], Luisa Frescura[8], Jose Antonio Izazola-Licea[8], Iris Semini[8], Peter Godfrey-Faussett[8], Paul R. De Lay[9], Adèle Schwartz Benzaken[10], Peter D. Ghys[8]

**1** Avenir Health, Glastonbury, Connecticut, United States of America, **2** Burnet Institute, Melbourne, Victoria, Australia, **3** East-West Center, Honolulu, Hawaii, United States of America, **4** MRC Centre for Global Infectious Disease Analysis, Imperial College London, London, United Kingdom, **5** Centre for Health Economics, University of York, York, United Kingdom, **6** Heidelberg Institute of Global Health, Heidelberg University, Heidelberg, Germany, **7** Institute for Global Health, University College London, London, United Kingdom, **8** UNAIDS, Geneva, Switzerland, **9** Consultant, Washington, DC, United States of America, **10** AIDS Health Care Foundation, Los Angeles, California, United States of America

* Jstover@AvenirHealth.org

**Data Availability Statement:** Model inputs and outputs are available in the supplemental appendix. The software can be downloaded from www.

## Abstract

### Background

UNAIDS has established new program targets for 2025 to achieve the goal of eliminating AIDS as a public health threat by 2030. This study reports on efforts to use mathematical models to estimate the impact of achieving those targets.

### Methods and findings

We simulated the impact of achieving the targets at country level using the Goals model, a mathematical simulation model of HIV epidemic dynamics that includes the impact of prevention and treatment interventions. For 77 high-burden countries, we fit the model to surveillance and survey data for 1970 to 2020 and then projected the impact of achieving the targets for the period 2019 to 2030. Results from these 77 countries were extrapolated to produce estimates for 96 others. Goals model results were checked by comparing against projections done with the Optima HIV model and the AIDS Epidemic Model (AEM) for selected countries. We included estimates of the impact of societal enablers (access to justice and law reform, stigma and discrimination elimination, and gender equality) and the impact of Coronavirus Disease 2019 (COVID-19). Results show that achieving the 2025 targets would reduce new annual infections by 83% (71% to 86% across regions) and AIDS-related deaths by 78% (67% to 81% across regions) by 2025 compared to 2010. Lack of progress on societal enablers could endanger these achievements and result in as many as 2.6 million (44%) cumulative additional new HIV infections and 440,000 (54%) more AIDS-related deaths between 2020 and 2030 compared to full achievement of all targets. COVID-19–related disruptions could increase new HIV infections and AIDS-related deaths by 10%

AvenirHealth.org. Individual country files are available at www.UNAIDS.org.

**Funding:** JS, RG, YT and TB received funding from the Bill & Melinda Gates Foundation (www.GatesFoundation.org) under grant OPP1191665. SK received funding from UNAIDS under 2019/05308. The Funders had no role in study design, data collection and analysis, decision to publish, or preparation of the manuscript.

**Competing interests:** The authors have declared that no competing interests exist.

**Abbreviations:** ABYM, adolescent boys and young men; AEM, AIDS Epidemic Model; AGYW, adolescent girls and young women; ART, antiretroviral therapy; ASM, Age-Structured Model; COVID-19, Coronavirus Disease 2019; CSE, comprehensive sexuality education; DHS, Demographic and Health Surveys; EE, economic empowerment; FSW, female sex worker; GATHER, Guidelines for Accurate and Transparent Health Estimates Reporting; GBV, gender-based violence; IBBS, Integrated Behavioral and Biomarker Surveys; IMR, Incidence-Mortality Ratio; IPR, Incidence-Prevalence Ratio; IPV, intimate partner violence; MSM, men who have sex with men; NSP, needle-syringe programs; OST, opioid substitution therapy; PEP, postexposure prophylaxis; PHIA, Population-Based HIV Impact Assessment; PLHIV, people living with HIV; PMTCT, prevent mother-to-child transmission; PrEP, pre-exposure prophylaxis; PWID, people who inject drugs; RSM, Risk-Structured Model; STI, sexually transmitted infection; SW, sex worker; TG, transgender people; VMMC, voluntary medical male circumcision.

in the next 2 years, but targets could still be achieved by 2025. Study limitations include the reliance on self-reports for most data on behaviors, the use of intervention effect sizes from published studies that may overstate intervention impacts outside of controlled study settings, and the use of proxy countries to estimate the impact in countries with fewer than 4,000 annual HIV infections.

## Conclusions

The new targets for 2025 build on the progress made since 2010 and represent ambitious short-term goals. Achieving these targets would bring us close to the goals of reducing new HIV infections and AIDS-related deaths by 90% between 2010 and 2030. By 2025, global new infections and AIDS deaths would drop to 4.4 and 3.9 per 100,000 population, and the number of people living with HIV (PLHIV) would be declining. There would be 32 million people on treatment, and they would need continuing support for their lifetime. Incidence for the total global population would be below 0.15% everywhere. The number of PLHIV would start declining by 2023.

## Author summary

### Why was this study done?

- The previous UNAIDS strategic plan expired in 2020, creating a need for a new plan to guide efforts for the next 5 years.

- Modeling contributed to the development of the new plan by assessing the epidemiological impact of proposed intervention coverage targets and estimating the financial resources needed to achieve them.

### What did the researchers do and find?

- We applied mathematical simulation models to 77 high HIV burden countries to examine the effects of the proposed intervention coverage targets on trends in new HIV infections and AIDS-related deaths. The results were extended to a total of 173 countries to provide a comprehensive global analysis.

- Results show that achieving the 2025 targets would reduce new annual infections by 83% (71% to 86% across regions) and AIDS-related deaths by 78% (67% to 81% across regions) by 2025 compared to 2010.

- Progress on societal enablers (access to justice, prevention of stigma and discrimination, and prevention of gender-based violence [GBV]) is essential to achieve these targets.

- Coronavirus Disease 2019 (COVID-19)–related disruptions could increase new HIV infections and AIDS-related deaths in the short term, but targets could still be achieved by 2025.

**What do these findings mean?**

- Although global efforts have failed to achieve the ambitious 2020 targets, it is still possible to achieve the 2030 goal of eliminating AIDS as a public health threat.

- Many of the 2025 intervention coverage targets have already been achieved in some settings. Additional effort is required to accelerate progress in all countries and in all populations.

## Introduction

As part of the Sustainable Development Goals, the UN has established a goal of ending AIDS as a public health threat by 2030 [1]. In 2014, UNAIDS released the Fast-Track strategy for achieving interim 2020 targets that would put the world on track to achieve the 2030 goal [2,3]. It included the 90–90–90 testing and treatment targets (90% of people living with HIV (PLHIV) know their status, 90% of those who know their status are on antiretroviral therapy (ART), and 90% of people receiving ART are virally suppressed); ambitious prevention targets for key populations, adolescent girls and young women (AGYW); and resources to address societal enablers and health system strengthening. Mathematical modeling was used to estimate the impact of achieving these targets on the global number of new HIV infections and AIDS-related deaths [3]. Results indicated that new infections and AIDS deaths could decline by nearly 90% from 2010 to 2030. Intermediate targets of 75% declines in each indicator by 2020 were also established. By 2020, significant progress had been made, although it fell short of these intermediate targets. By the end of 2020, 90% of those on ART were virally suppressed, but only 84% of PLHIV knew their status, and 87% of those were on treatment. By 2020, the annual global number of new infections had fallen by only 31% from 2010 and AIDS deaths by only 47% [4].

UNAIDS worked with a wide range of stakeholders to review past efforts [5] and propose a new set of programmatic targets for 2025 to speed progress toward achieving the 2030 goals [6]. Papers describing the process and background research that contributed to the strategy development are available in the *PLOS Medicine* Special Collection "UNAIDS HIV Targets" (https://collections.plos.org/collection/unaids-hiv). The purpose of this paper is to describe the modeling done to estimate the epidemiological impact of achieving these 2025 targets. The modeling contributed to the process in several ways including (1) translating the intervention coverage targets into impact on new infections and AIDS-related deaths to determine whether the targets were sufficiently ambitious to achieve the UNAIDS global goals of 90% reduction in both indicators from 2010 to 2030; (2) estimating the impact of the targets in each country and in each population group in order to assess the equity of the benefits; and (3) quantifying the numbers of people reached by each service in order to estimate the cost of achieving the targets.

## Methods

Our analysis plan was to assess the impact of the new HIV targets by applying mathematical simulation models to individual country data from 1970 to 2019 and then projecting to 2030 assuming intervention targets would be met. The information provided here follows the

Guidelines for Accurate and Transparent Health Estimates Reporting (GATHER) as documented in S1 GATHER Checklist.

Models were fit to 77 high-burden countries (Table 1), which accounted for 94% of global new HIV infections and 95% of AIDS deaths in 2019 [7]. Projections for the remaining 93 countries included as part of UNAIDS global reporting were produced using proxies by

**Table 1. Countries modeled with goals.**

| Modeled countries | Modeled countries |
|---|---|
| **Asia-Pacific region** | **Latin America and Caribbean** |
| Afghanistan | Brazil |
| Bangladesh | Colombia |
| Cambodia | Cuba |
| China | Guatemala |
| India | Haiti |
| Indonesia | Jamaica |
| Lao PDR | Mexico |
| Mongolia | Paraguay |
| Myanmar | **North Africa and Middle East** |
| Pakistan | Djibouti |
| Papua New Guinea | Lebanon |
| Philippines | Morocco |
| Tajikistan | Sudan |
| Thailand | **West and Central Africa** |
| Timor-Leste | Benin |
| Vietnam | Burkina Faso |
| **Eastern Europe and Central Asia** | Burundi |
| Azerbaijan | Cameroon |
| Kazakhstan | Central African Republic |
| Kyrgyzstan | Chad |
| Republic of Moldova | Congo |
| Russian Federation | Côte d'Ivoire |
| Ukraine | Democratic Republic of the Congo |
| **East and Southern Africa** | Equatorial Guinea |
| Angola | Gabon |
| Botswana | Gambia |
| Eritrea | Ghana |
| Ethiopia | Guinea |
| Kenya | Guinea-Bissau |
| Lesotho | Liberia |
| Malawi | Mali |
| Mozambique | Niger |
| Namibia | Nigeria |
| Rwanda | Senegal |
| South Africa | Sierra Leone |
| South Sudan | Togo |
| Swaziland | **West and Central Europe and North America** |
| Uganda | France |
| United Republic of Tanzania | Italy |
| Zambia | United Kingdom |
| Zimbabwe | United States of America |

applying trends for new infections and AIDS-related deaths from one of the modeled countries. Proxy countries were assigned to each nonmodeled country by choosing the modeled country from the same or neighboring region with the highest correlation in new infections between 1970 and 2020 with the nonmodeled country (Table 2).

**Table 2. Countries modeled using a proxy country.**

| Modeled countries | Serves as proxy for |
|---|---|
| **Asia-Pacific region** | |
| Cambodia | Nepal |
| China | Brunei Darussalam and DPR Korea |
| India | Australia and Republic of Korea |
| Myanmar | Japan, New Zealand, and Sri Lanka |
| Pakistan | Malaysia |
| Papua New Guinea | Maldives |
| Thailand | Singapore |
| Timor-Leste | Fiji |
| Vietnam | Bhutan |
| **Eastern Europe and Central Asia** | |
| Kazakhstan | Croatia, Cyprus, Czech Republic, Montenegro, and Serbia |
| Kyrgyzstan | Belarus and Uzbekistan |
| Republic of Moldova | Armenia |
| Russian Federation | Bosnia and Herzegovina, Bulgaria, Georgia, North Macedonia, and Turkmenistan |
| Ukraine | Albania |
| **East and Southern Africa** | |
| Eritrea | Mauritius and Somalia |
| South Sudan | Madagascar |
| Uganda | Comoros |
| **Latin America and Caribbean** | |
| Brazil | Chile and Honduras |
| Cuba | Argentina and Bolivia |
| Guatemala | Barbados, Dominican Republic, El Salvador, Suriname, and Trinidad and Tobago |
| Haiti | Belize and Venezuela |
| Jamaica | Bahamas, Guyana, and Nicaragua |
| Mexico | Costa Rica, Panama, and Uruguay |
| Paraguay | Ecuador and Peru |
| **North Africa and Middle East** | |
| Lebanon | Algeria, Bahrain, Egypt, Israel, Jordan, Kuwait, Oman, Qatar, Saudi Arabia, Tunisia, UAE, and Yemen |
| Morocco | Iran and Libya |
| Sudan | Syria |
| **West and Central Africa** | |
| Cameroon | Cape Verde |
| Chad | Mauritania |
| **West and Central Europe and North America** | |
| France | Iceland, Ireland, Lithuania, Luxembourg, Malta, Poland, Slovakia, and Turkey |
| Italy | Belgium, Canada, Denmark, Estonia, Greece, Latvia, Netherlands, Norway, Portugal, Romania, Spain, and Switzerland |
| United Kingdom | Austria, Hungary, and Sweden |
| United States of America | Finland, Germany, and Slovenia |

The main results presented here were produced using one of 2 different versions of the Goals model: for 39 countries in sub-Saharan Africa with generalized epidemics, where the age pattern of sexual contact drives the epidemic, the age-structured version of the model, Goals ASM (Age-Structured Model), was used, while for 38 countries from other regions with concentrated epidemics, characterized by HIV transmission among key populations and their partners, the risk structured version of the model, Goals RSM (Risk-Structured Model), was used. For each country, the model was used to project new infections and AIDS-related deaths under the assumption that all targets were achieved. Infections and deaths averted were calculated by comparing these results with a counterfactual scenario in which coverage of all interventions remained constant from 2019 to 2030. The contribution of individual interventions to the overall impact was estimated by scaling up each intervention one at a time, calculating the infections averted for each intervention, and then normalizing to sum to total infections averted with all interventions scaled up together.

Results from the Goals models were validated by comparing with outputs from 2 other models, Optima HIV, and the AIDS Epidemic Model (AEM), in 7 countries for which those models have been used for strategic planning. Descriptions of each model and the validations are given in the following sections.

## Goals RSM

The risk structured version of the Goals model calculates HIV incidence in the adult population between the ages of 15 and 49 using 6 categories for men (not yet sexually active, in a stable partnership, with multiple partners in the last year, clients of female sex workers (FSW), men who have sex with men (MSM), and people who inject drugs (PWID)) and 5 categories for women (not yet sexually active, in a stable partnership, with multiple partners in the last year, FSW, and PWID). (At this time, the model does not include a separate group for transgender people (TG) due to a lack of behavioral data. More data are becoming available, so we plan to add transgender population groups in the near future.) Each risk group is defined in terms of size and behaviors such as number of partners per year, acts per partner, condom use, and needle sharing. Transition between groups is based on average duration within each group. Partners are chosen from within the same risk group except for those in stable partnerships where partners can be from any risk group depending on marriage rates. HIV transmission is determined by the number of partners, the number of contacts per partner, the probability of encountering an infected partner, and the probability of transmission per act adjusted for partner's stage of infection, type of sex, presence of another sexually transmitted infection (STI) in either partner, effective ART use by the infected partner and condom use, male circumcision (voluntary medical male circumcision, VMMC), clean needles, and pre-exposure prophylaxis (PrEP) in the susceptible partner. Incidence among adults 15 to 49 is used to estimate incidence by age group (15 to 19 up to 80+) using incidence rate ratios by age that have previously been calculated by fitting prevalence by age to household surveys or from case reports of new diagnoses. New adult infections are tracked by CD4 count and ART status. AIDS-related mortality is determined by CD4 count, age, sex, and ART status. New child infections are determined from mother-to-child transmission and HIV-infected children also are followed by CD4 category, sex, age, and ART status. Full details of the model are provided elsewhere [8]. The model is implemented for an individual country by using country-specific data for demographic indicators (base year population, fertility, mortality, and migration), behavioral indicators (number and type of partners and condom use), and HIV program data (number of people on ART and number of women receiving prophylaxis to prevent mother-to-child transmission (PMTCT) and number of male circumcisions). The model is fit to data on prevalence from surveys, surveillance, and routine testing by varying the epidemiological

parameters within published ranges. The ranges used for the epidemiological parameters are given in S1 Table. The median values of the fitted parameters by county are provided in S2 Table. Ranges on the fitted values are used to generate uncertainty intervals on model output. The model is available for download free of charge from the Avenir Health website as a module in the Spectrum software at https://avenirhealth.org/software-spectrum.php.

## Goals ASM

The age structured Goals model, Goals ASM, represents HIV transmission driven by age-related factors, in contrast to the risk structured version that is driven by behavioral risk groups. Like Goals RSM above, Goals ASM uses Spectrum's cohort component projection method to simulate population dynamics and uses Spectrum's AIDS Impact Module to model HIV disease progression and mortality by age, sex, and CD4 cell count and to track ART status and simulate mother-to-child transmission. Goals ASM is designed to model generalized HIV epidemic contexts and represents heterosexual HIV transmission based on age-dependent inputs: rates of partner change, preferential sexual mixing, and the risk of HIV transmission within heterosexual serodiscordant partnerships. These transmission risks depend on condom use within the partnership; the HIV infection stage, ART status, and viral suppression status of the partner living with HIV; and male circumcision status, use of PrEP methods, and STI status of the HIV susceptible partner. The model incorporates general population behavior change programs, including economic empowerment (EE) and school-based prevention and sexuality education programs. The impacts of these programs are mediated by their coverage levels and their effects on frequency of condomless sex and other risk behaviors.

MSM, FSW, and PWID bear high HIV risk even in settings with generalized HIV epidemics [9,10], but critical data like population sizes and HIV burden estimates, when available, are often sparse in these settings [11–13]. Given these limitations, Goals ASM approximates the impact of key population interventions based on the proportion of men who are MSM, women who are FSW, and adults who inject drugs; HIV incidence in key populations relative to the general population; intervention coverages; and the reduction in HIV incidence among people reached by interventions.

Goals ASM is implemented for an individual country using country-specific data for demographic indicators, behavioral indicators, and HIV program data. The model is fitted to data on HIV prevalence by age from nationally representative household surveys and from surveillance and routine testing of pregnant women during antenatal care. S1 Text provides details of the HIV transmission model and fitting methods. The model is available for download free of charge from the Avenir Health website as a module in the Spectrum software at https://avenirhealth.org/software-spectrum.php.

## Optima HIV

The population-based compartmental Optima HIV model represents HIV transmission driven by age- and risk-related factors. The model was implemented for individual countries. Models were informed using data and estimates for demographic indicators (annual population size for each population group reflecting fertility, migration, and background mortality), epidemiological parameters including probability of transmission per sex act, variation by stage of infection (informed by CD4 cell counts and viral load monitoring), HIV testing rate, presence of other ulcerative STIs and/or tuberculosis, and effectiveness of condoms, circumcision, and nonsuppressive or suppressive ART, and mortality rate, behavioral parameters including number and type of partners (regular, casual, or commercial sexual; injecting), sex acts per partner, condom use, and needle sharing, differing by age and risk group (including

FSW, clients of sex workers (SWs), MSM, and PWID), and HIV program data (number of people on ART, number of women receiving prophylaxis to PMTCT, and proportion of males who are circumcised). Parameters were varied to fit the models to country-specific prevalence estimates from surveys, surveillance, and routine testing. HIV acquisition depends on characteristics including type and number of acts, circumcision and PMTCT status, and PrEP and PEP use, and population status (stage of infection, HIV testing, diagnosis, HIV prevalence, and nonsuppressive or suppressive ART use) in specified partnerships. Risk of HIV-related mortality is determined by dynamically changing CD4 cell count depending on treatment status (untreated or treated with suppressive or nonsuppressive ART). Full details of the model are provided elsewhere [14]. Model parameters and corresponding data sources are provided in the Optima HIV User Guide volume 6 at http://optimamodel.com/parameter-data-sources. The Optima HIV model is available free of charge from http://hiv.optimamodel.com/.

### AIDS epidemic model

The AEM was developed in the concentrated epidemics of Asia. It is a risk structured model built around key populations: FSW and clients, MSM, PWID, transgenders, and the remaining non-key population of men and women. Each group is incorporated in the model as a set of HIV+ and HIV− compartments containing all individuals 15 and older meeting the group's characteristics. People enter at age 15, and subsequent movement is allowed between the groups based on average durations of group membership. While there is no further age structure, the HIV+ groups are subdivided into the on and off ART CD4 groups that define the Spectrum CD4 model, which AEM uses for its mortality calculations [15]. AEM calculates new infections in each group based on frequency of sexual and injecting risk behaviors with different types of partners, levels of protective behavior (e.g., condom use and clean needle use), size as a percent of 15 to 49 population, and HIV and STI prevalence over time. The number on ART can be specified by sex and apportioned by ART need or specified separately for each population. Any of these inputs can vary over time. Adjustments to a set of transmission probabilities (vaginal male to female, vaginal female to male, anal insertive, anal receptive, and needle sharing), cofactors (STI, circumcision, and primary infection), and start years for components of the epidemic (heterosexual, PWID, and MSM) are made to obtain a fit between the prevalent infections calculated by applying AIDS and non-AIDS deaths to new infections over time and the observed prevalence in each included population with prevalence inputs. For application in-country, the various inputs are extracted from critical review of published articles, gray literature, epidemiological and behavioral data systems, and program data systems [16]. The country team of technical experts that implements the model then fits calculated and observed prevalence trends in key populations to produce a model tuned to the country's unique history and situation. AEM does not have a separate pediatric component, but instead can be used as an incidence source in Spectrum where pediatric calculations can be carried out. The AEM model and its use for the impact analyses discussed later are described in more detail in S2 Text.

### Program targets

The program targets in the new UNAIDS plan build on the previous ones by establishing targets for 2025 and by differentiating targets by risk of infection. The targets have been expanded to include a more comprehensive plan for addressing stigma and discrimination, criminalization of certain behaviors, and gender-based violence (GBV). Treatment targets are to be achieved in all relevant populations, notably age and sex, key populations, geography, migrant status, and other factors that may relate to inequalities.

**Table 3. 2025 targets for key populations.**

| Intervention | SWs | MSM | TG | Prisoners and others in closed settings | PWID | Applies to |
|---|---|---|---|---|---|---|
| Condoms/lube | 90% | 95% | 95% | 90% | 95% | Use at last sex by people not taking PrEP and who have nonregular partner whose HIV viral load status is not known to be undetectable |
| PrEP | | | | | | Uninfected population |
| Very high risk | 80% | 50% | 50% | 15% | 15% | |
| High risk | 15% | 15% | 15% | 5% | 5% | |
| Low risk | 0% | 0% | 0% | 0% | 0% | |
| Sterile needles and syringes | | | | 90% | 90% | PWID |
| OST | | | | | 50% | People who are opioid dependent |
| STI screening and treatment | 80% | 80% | 80% | | | People with symptoms of STIs |
| Appropriate health or community-led services | 90% | 90% | 90% | 100% | 90% | All |
| PEP (nonoccupational exposure) | 90% | 90% | 90% | 90% | 90% | Those with recent exposure |
| Knowledge of status | 95% | 95% | 95% | 95% | 95% | PLHIV |
| On ART | 95% | 95% | 95% | 95% | 95% | PLHIV who know their status |
| Virally suppressed | 95% | 95% | 95% | 95% | 95% | PLHIV on ART |

ART, antiretroviral therapy; MSM, men who have sex with men; OST, opioid substitution therapy; PEP, postexposure prophylaxis; PLHIV, people living with HIV; PrEP, pre-exposure prophylaxis; PWID, people who inject drugs; STI, sexually transmitted infection; SW, sex worker; TG, transgender people.

Prevention targets are defined based on geographic risk or behavioral risk or both. Areas of geographic risk are defined as those areas having HIV incidence in the target population above 3% (very high risk), 0.3% to 3% (high risk), or less than 0.3% (low risk). HIV programs for SWs are categorized by national prevalence, while those for MSM, TG and prisoners are categorized by incidence within those populations. For PWID, high-risk settings are considered to be those with low coverage of needle-syringe programs (NSP) and opioid substitution therapy (OST), medium-risk settings are those with some NSP and OST, and low risk are those with high coverage of NSP and OST programs and adequate provision of syringes and needles. The targets for key populations are shown in Table 3.

For AGYW, adolescent boys and young men (ABYM) and adult adults aged 25 and older, targets for postexposure prophylaxis (PEP), EE, and VMMC are based on 4 levels of incidence for each population at the district level: very high (>3%), high (1% to 3%), moderate (0.3% to 1%), and low (<0.3%). For condoms, PrEP, STI screening and treatment, and comprehensive sexuality education (CSE) in schools, 3 risk strata are defined: high and very high (incidence of >3% or incidence of 1% to 3% and reported high-risk behavior), moderate (incidence of 0.3% to 1% and reported high-risk behavior or incidence of 1% to 3% and no reported high-risk behavior) and low (incidence of <0.3% or incidence 0.3% to 1% and reported no reported high-risk behavior). High-risk behaviors are reporting 2 or more partners in the last year or an episode of an STI. The targets for the general population are shown in Table 4.

## Data sources

Demographic data (population by age and sex and rates of fertility, mortality, and migration) were taken from World Population Prospects 2019 [17]. Population sizes for key populations were based on the UNAIDS Key Population Atlas (kpatlas.unaids.org/dashboard). For countries without data, we applied regional averages of the percentage of the relevant population. Information on reported behaviors (multiple partners and an episode of an STI in the last

**Table 4. 2025 targets for general populations.**

| Intervention | Target by strata | Applies to |
|---|---|---|
| Condoms | Very high: 95% | Use at last sex by people not taking PrEP and who have nonregular partner whose HIV viral load status is not known to be undetectable |
| | Moderate: 70% | |
| | Low: 50% | |
| PrEP use | Very high: 50% | All HIV–negative sexually active adults |
| | Moderate: 5% | |
| | Low: 0% | |
| STI screening and treatment | Very high: 80% | All sexually active adults with STI symptoms |
| | Moderate: 10% | |
| | Low: 10% | |
| CSE in school | Very high: 90% | All males and females enrolled in secondary education |
| | Moderate: 90% | |
| | Low: 90% | |
| EE | Very high: 20% | AGYW |
| | Moderate: 20% | |
| | Low: 0% | |
| PEP (nonoccupational exposure) | Very high: 90% | All adults with recent exposure to HIV |
| | High: 50% | |
| | Moderate: 5% | |
| | Low: 0% | |
| PEP (nosocomial) | Very high: 90% | All adults with recent nosocomial exposure to HIV |
| | High: 80% | |
| | Moderate: 70% | |
| | Low: 50% | |
| VMMC | 90% | ABYM (15–24) and men aged 25–49 in 15 priority countries |
| Knowledge of status | 95% | All PLHIV |
| On ART | 95% | All known PLHIV |
| Viral suppression | 95% | All those on ART |
| PMTCT | 95% | All HIV+ pregnant women |

ABYM, adolescent boys and young men; AGYW, adolescent girls and young women; ART, antiretroviral therapy; CSE, comprehensive sexuality education; EE, economic empowerment; PEP, postexposure prophylaxis; PLHIV, people living with HIV; PMTCT, prevent mother-to-child transmission; PrEP, pre-exposure prophylaxis; STI, sexually transmitted infection; VMMC, voluntary medical male circumcision.

year) was from Demographic and Health Surveys (DHS; https://dhsprogram.com/) where available, with regional averages used for countries without surveys. For each country, the percentage of AGYW, ABYM, and adults 25+ living in districts with very high, high, or moderate incidence was based on official estimates (https://aidsinfo.unaids.org/) produced by national

HIV estimates teams using the Naomi geospatial model informed with survey and program data [18]. These district level estimates are only available for 25 countries in sub-Saharan Africa. Therefore, we assumed that all AGYW, ABYM, and adults 25+ in other countries were in the low-risk category. Intervention coverage for 2019 was based on values from the most recent year reported to UNAIDS (https://aidsinfo.unaids.org/). Country models were fit to prevalence data. For general populations, prevalence data were from DHS and Population-Based HIV Impact Assessment (PHIA) surveys (https://phia.icap.columbia.edu/). Prevalence among key populations was from Integrated Behavioral and Biomarker Surveys (IBBS) as reported in the UNAIDS Key Population Atlas (https://kpatlas.unaids.org/). Impacts of bio-medical interventions (ART, condoms, PrEP, NSEP, OST, and VMMC) were based on the probability of transmission per sexual act or unsafe injection sourced from published studies. Impacts of behavior change interventions (services for key populations, CSE, and EE) for key behaviors (condom use, number of partners, age at first sex, and needle sharing) are based on impact studies. Impact values and sources are provided in S3 Table.

## Uncertainty

Uncertainty in the projections derives largely from ranges around current estimates of new infections and AIDS-related deaths as a result of uncertainty associated with prevalence from surveys and surveillance data and in progression and mortality by CD4 count. Since the projections are based on achieving predefined coverage targets, there is no uncertainty in future coverage or in the impact of ART (since viral suppression is a target). Uncertainty associated with the impact of primary prevention interventions is small since coverage targets are high and prevention interventions interact so that less impact of one intervention is offset by increased impacts of others.

## Impact of societal enablers

A special challenge was to estimate the impact of progress in societal enablers. While a broad mix of societal factors affect vulnerability to HIV, this analysis focused on 3 for which data were available: access to justice and law reform to lift punitive and criminalizing laws, elimination of HIV stigma and discrimination, and gender equality. Criminalization of sex work, same sex intercourse, and drug use; stigma and discrimination; and GBV can result in low use of prevention, testing and treatment services, as discussed below. Interventions to address these issues can lead to more utilization of services. However, since the 2025 prevention and treatment targets already specify very high coverage of all services, it is not possible to demonstrate how progress on societal enablers would lead to better outcomes. Therefore, we assumed that a favorable enabling environment was essential to achieving the 2025 programmatic targets and modeled how the lack of additional progress on societal enablers would lead to shortfalls in achieving the programmatic targets and, therefore, more HIV infections and deaths. This section describes how we estimated the impact of these 3 societal enablers.

**Stigma and discrimination.** Reduction and elimination of HIV-related stigma and discrimination (i.e., directed to PLHIV or to key populations at risk of HIV or held by service providers) refers to at least 3 different manifestations: community level discrimination, healthcare provider discrimination, and internalized stigma. Studies have measured the effects of internalized stigma on access to care and treatment and found that it leads to reduced likelihood of testing for HIV [19], late linkage to care [20,21], lower levels of adherence to treatment [22], and lower levels of viral suppression among those on treatment [23]. We have used these studies to estimate the effects of internalized stigma on the treatment cascade: knowledge of status on treatment and viral suppression. If we assume that the global goals of 95–95–95 for

**Table 5. Estimation of treatment cascade in absence of progress on stigma.**

| Cascade component | Study | Indicator | Adjusted odds ratio | Odds of 95% | Odds with stigma | Percentage achievement with stigma |
|---|---|---|---|---|---|---|
| Testing | Golub and Gamarel [19] | Likelihood of testing | 0.54 | 19 | 10.3 | 0.91 |
| Linkage | Sabapathy and colleagues [20] | Late linkage to care | 1.71 to 1.82 | 19 | 10.4 to 11.11 | 0.91 to 0.92 |
| | Gesesew and colleagues [21] | Late presentation to care | 2.4 (1.6 to 3.6) | 19 | 7.9 | 0.84 to 0.92 |
| Adherence | Katz and colleagues [22] | Nonadherence | 1.74 | 19 | 10.9 | 0.92 |
| | Hargreaves and colleagues [23] | Viral suppression | 0.83 | 19 | 15.8 | 0.94 |

Note: The achievement with stigma is calculated from the odds of each cascade component at its target value of 95% (odds of 95% = 19) multiplied by the adjusted odds ratio for positive improvements in testing or viral suppression or divided by the odds ratio for the negative effects of late linkage and nonadherence.

all relevant population groups can be achieved only in the absence of internalized stigma, then for those with internalized stigma, the maximum achievements would be 91% knowledge of status (because of reluctance to test), 84% to 92% of those knowing their status on ART (because of poor linkage to care and high dropout), and 92% to 94% of those on ART that are virally suppressed (because of poor adherence) as shown in Table 5. In other words, instead of achieving the treatment cascade targets of 95–95–95, it might only be possible to achieve 91–88–93 without addressing stigma. While these estimates are based on few studies, only address one aspect of stigma and discrimination, and refer to late linkage to care rather than never linking, they are used here to illustrate the magnitude of the impact that might be expected with attention to stigma and discrimination.

These lower cascade values would affect the 22% of PLHIV with internalized stigma [22]. We ran the Goals model for each of 77 countries with these lower cascade targets to estimate the effects of not addressing stigma and discrimination.

**Access to justice.** The legal framework in a country can affect HIV prevalence in key populations [24]. The impact of decriminalization has been addressed in a study that modeled the effects of decriminalization of sex work and found about a 40% reduction in new infections among SWs over a 10-year period in Vancouver, Canada and Mombasa and Bellary, India [25]. For PWID, modeling has shown that decriminalization in Mexico coupled with OST could prevent 21% of new infections [26]. Data from the UNAIDS Key Population Atlas [27] indicate that 34% of 192 countries have laws that criminalize same sex sexual activity and 80% of 134 countries criminalize sex work. We estimated the effects of not achieving decriminalization targets by applying the reductions in new infections found in the above studies to countries that currently criminalize sex work and drug injection.

**Gender equality.** Gender equality is a broad topic that includes societal norms that place girls and women at increased risk of HIV. For this analysis, we focused on GBV as an important and signal component of gender equity for which some data exist. Data on the extent of GBV are available from national surveys, including DHS. GBV can lead to more unprotected sex, increased prevalence of other STIs, reduced testing, and reduced adherence to treatment and biomedical prevention. Studies have measured these relationships in a variety of settings [28]. The results were mixed but generally supported the idea that women subject to GBV were less likely to link to HIV care and less likely to adhere to treatment. Results on testing were mixed with some studies showing increased testing and others the opposite. The research is less clear on whether interventions to reduce GBV would lead to less risky behavior or whether perpetration of GBV is associated with risk of HIV acquisition that would persist even

if the violence ended. Studies have reported an association between violence and HIV infections in South Africa [29,30], reporting an incidence rate ratio of 1.51 (1.04 to 2.21) for the effect of intimate partner violence (IPV) on HIV incidence and on ART uptake in Zambia [31]. An analysis of the impact of scaling up programs to prevent IPV by UNFPA estimated that a global program to scale up prevention services could avert 14% of IPV cases by 2025 and 29% by 2030 [32]. With about one-third of women experiencing intimate or nonpartner violence [33], this implies that a global program to prevent IPV might avert about 5% of new HIV infections by 2030.

## COVID-19 disruptions

Disruptions in health services have the potential to lead to excess numbers of new HIV infections and AIDS-related deaths [34]. Based on data collected by UNAIDS on monthly services disruptions during 2020 [5], we examined the potential effects of disruptions on the impact of these targets by modeling 3 Coronavirus Disease 2019 (COVID-19) scenarios with disruptions starting in April 2020 and lasting 3 months, 6 months, or 2 years. During the disruption we, assumed that the rate of increase in ART coverage would be half of the pre-COVID-19 rate, no new VMMC during this time, 20% reduction in PMTCT services, and no PrEP scale-up.

## Results

The global and regional impacts of achieving the 2025 targets are shown in Fig 1 for new HIV infections and Fig 2 for AIDS-related deaths. New infections are estimated to have fallen by 31% from 2010 to 2020 [4]. Achieving the 2025 targets would result in a decline from 2010 of 83% by 2025. The declines by 2025 would be similar by region with a low of 71% in Western and Central Europe and North America and a high of 86% in East and Southern Africa. By 2025, there would be just 370,000 (250,000 to 490,000) new infections annually. The decline for AIDS deaths has been larger from 2010 to 2019 (47%) and would reach about 78% by 2025.

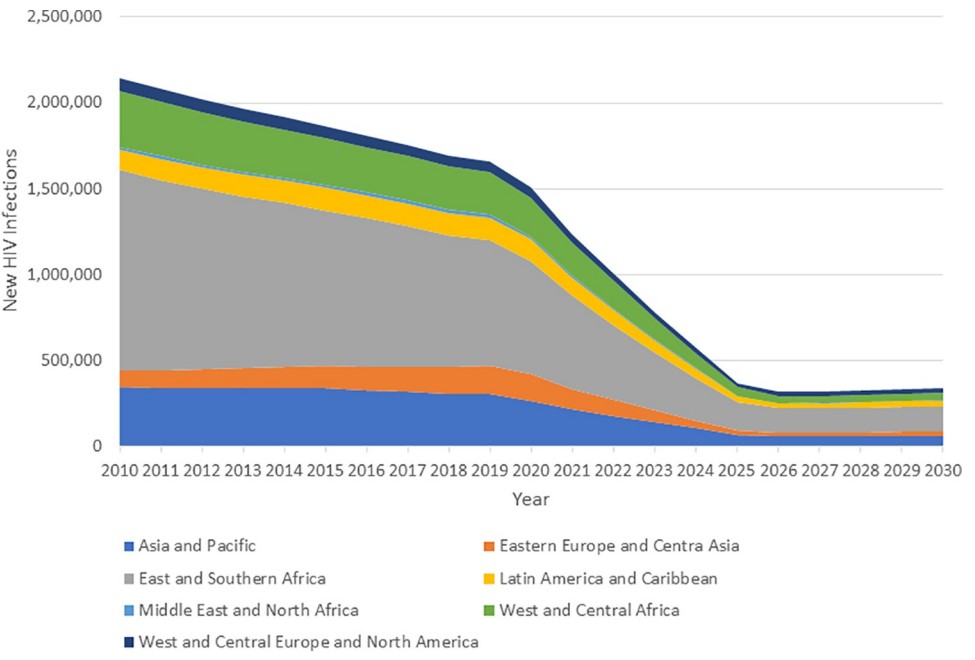

**Fig 1. New HIV infections from 2010 to 2019 and projection to 2030 if targets are achieved.**

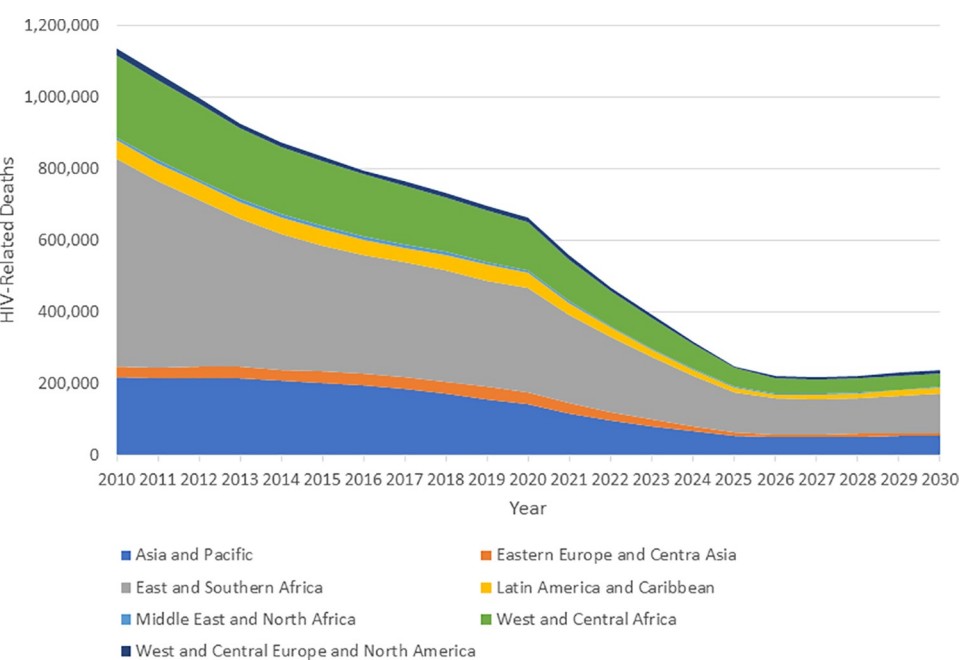

**Fig 2. AIDS deaths if 2025 targets are achieved.**

By 2025, there would be 250,000 (175,000 to 370,00) AIDS deaths annually. Results by region and year for all indicators are given in S4 Table.

Although the number of new infections exceeds the number of AIDS deaths in 2025, the number of PLHIV will be declining because of the additional non-AIDS–related deaths in PLHIV. As new infections fall and high ART coverage keeps people alive, the average age of the population living with HIV will rise, leading to a higher rate of non-AIDS–related mortality. If the targets are achieved, the number of PLHIV will decline by 1.7 million from a peak of 38.8 million in 2023 to 37.1 million by 2030.

New HIV infections among children have declined by 52% from 2010 to 2020, more than for the general population and, if these targets are achieved, will decline by 93% by 2025. In 2025, the mother-to-child transmission rate would decline to 1.7%.

New infections for AGYW have declined by 37% from 2010 to 2020 (somewhat faster than the total population) and would continue declining to reach 86% reduction by 2025. This implies that there would still be about 61,000 new infections in 2025. The decline is largest in Eastern and Southern Africa (88%) where 65% of new infections among AGYW are located and where PrEP and EE are especially targeted. The decline is just 23% in all other regions.

The rapid declines in new infections would put the world on a path to epidemic transition [35] as illustrated in Fig 3. The incidence mortality ratio is the ratio of new infections to deaths from all causes to PLHIV. When this ratio drops below 1.0, the number of PLHIV will be declining. If the targets are achieved, this threshold will be crossed in 2023. The incidence prevalence ratio is the ratio of new HIV infections to PLHIV. When this ratio is below 0.03, the number of PLHIV will eventually decline. This threshold will be crossed in 2022.

All interventions contribute to the impact of achieving the targets, but some may have more impact than others depending on the difference between current and target coverage, the percentage of new infections in the population group, the intervention targets, and the intervention effectiveness. We estimated the relative contribution to the total impact for 4 intervention packages: behavior change (condoms for the general population, CSE, and EE),

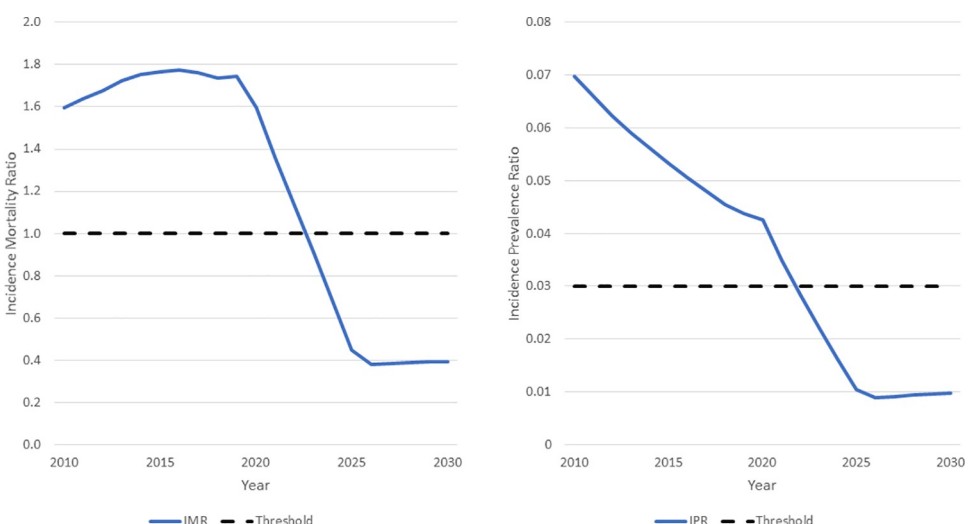

**Fig 3. Indicators of transition control.** IMR, incidence mortality ratio; IPR, incidence prevalence ratio.

key population prevention services (prevention services for key populations), biomedical prevention (VMMC and PrEP for adolescents and adults), and treatment (ART). Across all countries, treatment scale-up accounts for two-thirds of infections averted, key population services 17%, behavior change 14%, and biomedical 2%. This pattern varies by region as shown in Fig 4.

Achieving these targets implies that the number of people on ART would increase from 26 million in mid-2020 to 35 million by 2025 before declining slowly to 34 million by 2030.

Modeling of the COVID-19 scenarios indicates that the disruptions could lead to 10% more new infections in 2021 and 9% additional AIDS deaths, but, by 2025, the effect of the disruption would no longer be evident (S1 Fig).

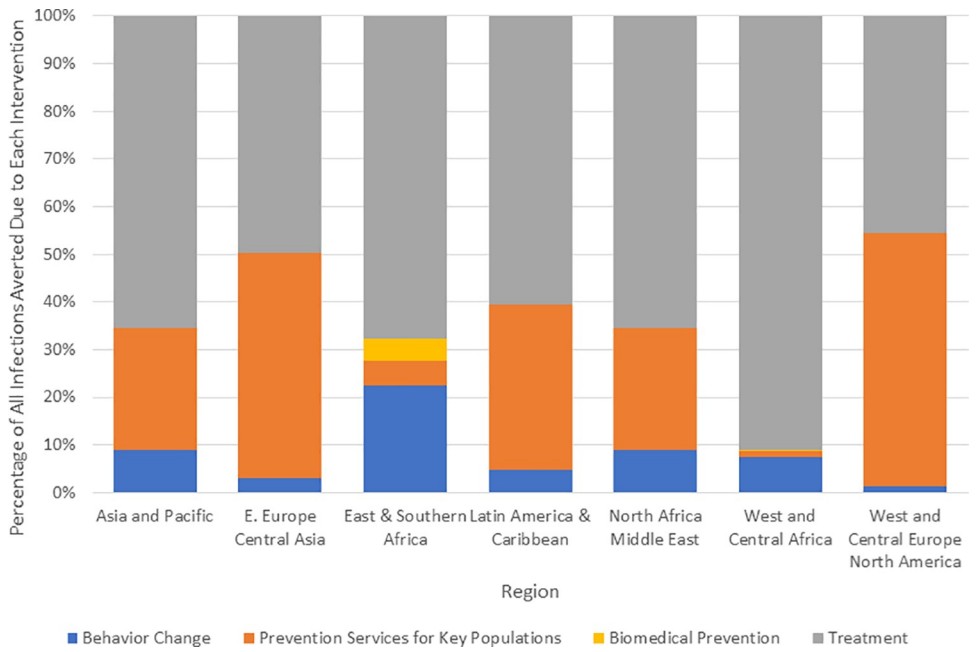

**Fig 4. Contribution to reduction in new HIV infections by intervention category.**

Reducing stigma and discrimination, improving access to justice and lifting punitive and criminalizing laws, and enhancing gender equality will be important to achieving these goals. Without such progress, we estimate that cumulative new infections between 2020 and 2030 would be about 44% higher than if the targets are achieved. There would be some 2 million additional infections due to falling short of the treatment targets, 750,000 due to criminalization of key population behaviors, and 70,000 due to lack of progress on preventing GBV (S2 Fig). Cumulative AIDS-related deaths would be 1.6 million (54%) higher in the absence of progress in these key areas.

The projected number of new HIV infections and AIDS-related deaths if the targets are achieved were compared between the Optima HIV and Goals model for 4 example countries (Eswatini, Malawi, Sudan, and Zimbabwe) for which Optima HIV had been previously applied. Fig 5 shows that the results are generally similar. (Optima shows higher mortality for Eswatini in the historical period, but the results are similar for the period 2019 to 2030.)

As additional validation of the Goals RSM for concentrated epidemic settings, the 2025 target setting exercise was replicated in 3 countries that use AEM: Cambodia, Indonesia, and Myanmar. Using the expected key population intervention behavioral impacts developed by the estimates teams in each country in their 2019 national AEM modeling work, the 2025 targets were applied using the AEM Intervention Workbook for each country, as described in more detail in S2 Text.

The impacts of the target scenario relative to the baseline for Indonesia and Myanmar are quite similar with both Goals and AEM (Fig 6).

## Discussion

The new targets for 2025 build on the progress made since 2010 and represent ambitious short-term goals. We have examined the impact of achieving these targets using mathematical models. The results show that achieving these targets could bring us close to the targets of reducing new HIV infections and AIDS-related deaths by 90% between 2010 and 2030. By 2025, global new infections and AIDS deaths would drop to 4.4 and 3.9 per 100,000 population, and the number of PLHIV would be declining. There would be 32 million people on treatment, and they would need continuing support for their lifetime. Incidence for the total global population would be below 0.15% everywhere. The number of PLHIV would decline by 1.7 million from a peak of 38.8 million in 2023 to 37.1 million by 2030.

Modeling has been used in the development of previous plans. Since those plans are intended to guide the response rather than predict the future, it is not possible to judge the accuracy of previous modeling since program implementation has lagged behind proposed targets. However, this round of modeling includes a number of improvements. The available data used to parameterize and fit models have improved significantly in the last few years with the implementation of many new national surveys (both PHIA and DHS), more national studies of key population size and behaviors, and increased availability of routine program data on testing, treatment access, and viral suppression. New tools to prepare HIV estimates at subnational levels in many countries allow targets to be differentiated by subnational geography. Finally, new studies on the effects of societal enablers on behaviors and uptake of services have enabled us, for the first time, to quantify the impact of progress in society enablers on HIV infections and deaths.

There are limitations in any modeling exercise. Some model inputs, such as sexual behavior, are based on self-reports that may be biased. To address this, models are fit to survey and surveillance data, but these sources are limited for some countries. Intervention effect sizes rely on published studies that may not be typical of real-world use and may not apply equally to all countries. The Goals models use a fixed number of populations groups that may not

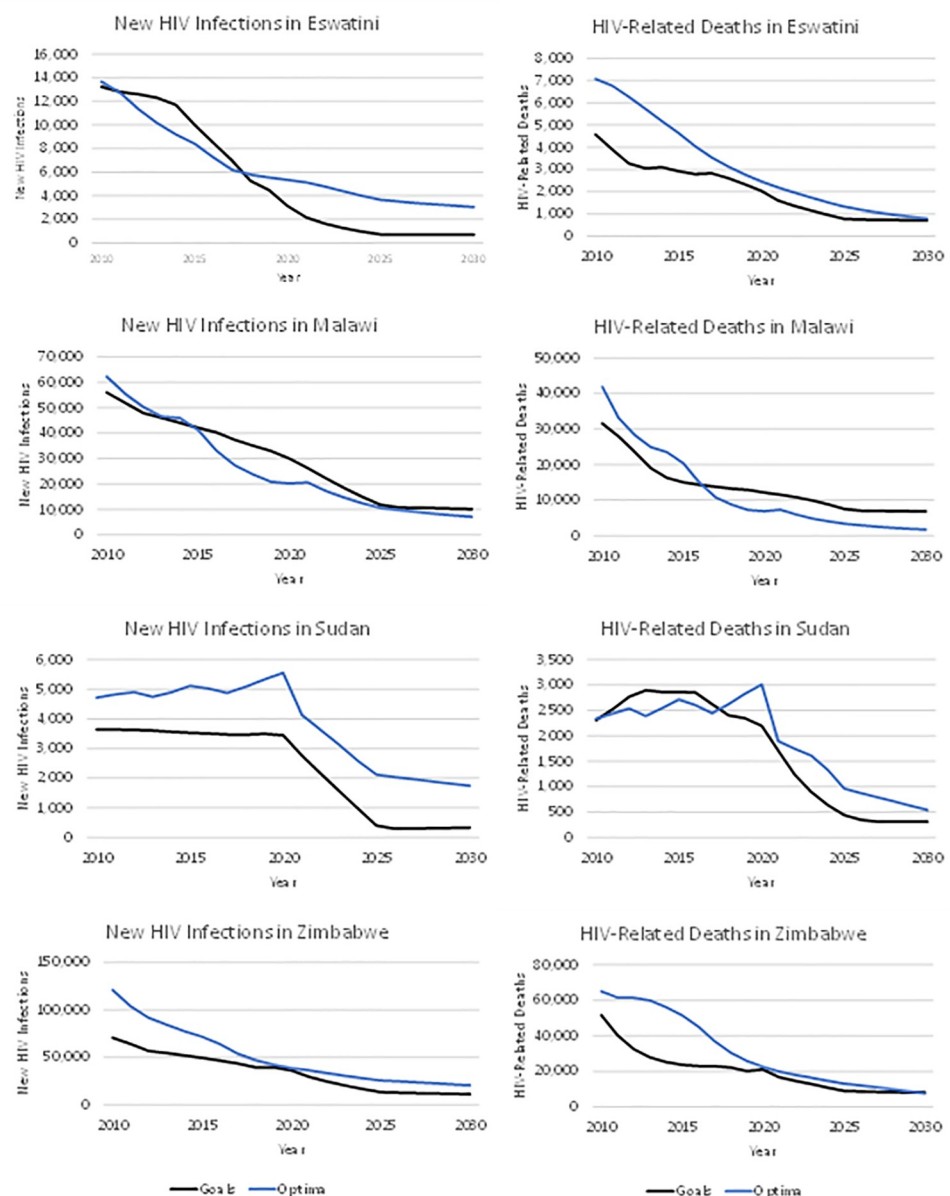

**Fig 5. Comparison of Optima HIV and goals projections.**

adequately capture the heterogeneity of risk behaviors within each group. The models assume that those within a population group who are not reached with an intervention have the same behavior as those who are reached, but we have limited data to support this assumption. Different models and different modeling teams may produce different results even when using the same data. We tried to address these limitations by comparing these results with those from 2 other models, Optima HIV and AEM, for a selection of countries. There were some differences in models results for the historical period in estimates of trends in mortality that were mostly due to assumptions about non-AIDS mortality and the effects of ART on CD4 counts. Differences in AIDS-related deaths are primarily attributable to differences in the background mortality between Goals and AEM that have been adjusted for in the most recent version of AEM. For Cambodia, Goals has a higher initial rate of new infections in the baseline, producing

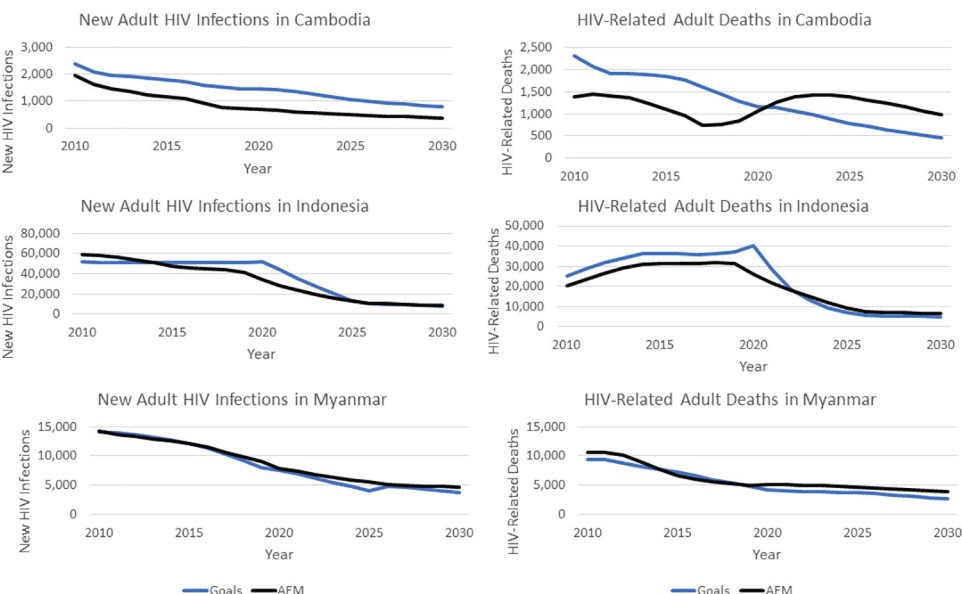

**Fig 6. Comparison of AEM and goals results.** AEM, AIDS Epidemic Model.

a flatter epidemic. However, the impact of the targeted interventions on new infections is similar, reducing them by about 45% between 2020 and 2030. The ART differences in Cambodia result because AEM, with its lower rate of new infections, produces an epidemic in decline. In AEM, because by design people cannot be removed from ART except through mortality, this already puts the AEM baseline on track to exceed the 2025 ART target and achieve the 90% target in 2030. Thus, there is little difference in the number on ART in the 2 AEM scenarios and AIDS-related deaths are similar. Overall, the models were in broad agreement on the reductions in new infections and AIDS deaths that would result if these targets can be achieved.

A global program to reduce stigma would include interventions to address internalized stigma, healthcare worker discrimination, and community norms. This analysis is focused only on internalized stigma, so it might underestimate the impact of a full program unless internalized stigma is a good indicator of all forms of stigma and discrimination.

Achieving these targets will require tremendous efforts by all involved to scale up treatment for all PLHIV and effective prevention measures for populations who most need them and to improve social conditions to remove barriers to progress. These targets are ambitious but not impossible. They have already been achieved in some countries and in some populations. The task ahead is to spread that success everywhere. This analysis indicates that the benefits would be considerable and well worth the effort.

## Supporting information

**S1 GATHER Checklist. The GATHER checklist.** GATHER, Guidelines for Accurate and Transparent Health Estimates Reporting.
(DOCX)

**S1 Text. Description of the Goals ASM model.** ASM, Age-Structured Model.
(DOCX)

**S2 Text. Description of the AEM model.** AEM, AIDS Epidemic Model.
(DOCX)

**S1 Table. Ranges for epidemiological parameters used for model fitting.**
(DOCX)

**S2 Table. Fitted parameter values by country.**
(DOCX)

**S3 Table. Effectiveness of HIV prevention interventions in the Goals model.**
(DOCX)

**S4 Table. Results by indicator, region, and year.**
(XLSX)

**S1 Fig. Effect of disruptions due to COVID-19 on new HIV infections and HIV-related deaths.** COVID-19, Coronavirus Disease 2019.
(TIF)

**S2 Fig. Impact of societal enablers on new HIV infections and HIV-related deaths.**
(TIF)

## Author Contributions

**Conceptualization:** John Stover, Paul R. De Lay, Peter D. Ghys.

**Data curation:** John Stover, Robert Glaubius, Yu Teng, Sherrie Kelly, Tim Brown.

**Formal analysis:** John Stover, Robert Glaubius, Yu Teng, Sherrie Kelly, Tim Brown.

**Funding acquisition:** John Stover, Peter D. Ghys.

**Investigation:** John Stover.

**Methodology:** John Stover, Robert Glaubius, Yu Teng, Sherrie Kelly, Tim Brown, Timothy B. Hallett, Paul Revill, Till Bärnighausen, Andrew N. Phillips, Christopher Fontaine, Luisa Frescura, Jose Antonio Izazola-Licea, Iris Semini, Peter Godfrey-Faussett, Paul R. De Lay, Adèle Schwartz Benzaken, Peter D. Ghys.

**Project administration:** John Stover, Peter D. Ghys.

**Resources:** John Stover, Peter D. Ghys.

**Software:** John Stover, Robert Glaubius, Yu Teng, Tim Brown.

**Supervision:** John Stover, Paul R. De Lay, Peter D. Ghys.

**Validation:** John Stover, Robert Glaubius, Yu Teng, Sherrie Kelly, Tim Brown, Timothy B. Hallett, Paul Revill, Till Bärnighausen, Andrew N. Phillips, Christopher Fontaine, Luisa Frescura, Jose Antonio Izazola-Licea, Iris Semini, Peter Godfrey-Faussett, Paul R. De Lay, Adèle Schwartz Benzaken, Peter D. Ghys.

**Visualization:** John Stover, Robert Glaubius, Yu Teng, Sherrie Kelly, Tim Brown, Paul R. De Lay, Adèle Schwartz Benzaken, Peter D. Ghys.

**Writing – original draft:** John Stover, Sherrie Kelly, Tim Brown, Peter D. Ghys.

**Writing – review & editing:** John Stover, Robert Glaubius, Yu Teng, Sherrie Kelly, Tim Brown, Timothy B. Hallett, Paul Revill, Till Bärnighausen, Andrew N. Phillips, Christopher Fontaine, Luisa Frescura, Jose Antonio Izazola-Licea, Iris Semini, Peter Godfrey-Faussett, Paul R. De Lay, Adèle Schwartz Benzaken, Peter D. Ghys.

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
