## [Editor Report · Decision Letter 0]

21 May 2021

Dear Dr Stover, 

Thank you for submitting your manuscript entitled "The Epidemiological Impact of the UNAIDS 2025 Targets for HIV" for consideration by PLOS Medicine.

Your manuscript has now been evaluated by the PLOS Medicine editorial staff and I am writing to let you know that we would like to send your submission out for external assessment.

However, before we can send your manuscript for assessment, we need you to complete your submission by providing the metadata that is required for full assessment. To this end, please login to Editorial Manager where you will find the paper in the 'Submissions Needing Revisions' folder on your homepage. Please click 'Revise Submission' from the Action Links and complete all additional questions in the submission questionnaire.

Please also substitute a structured abstract in PLOS Medicine style.

Please re-submit your manuscript within two working days, i.e. by May 25 2021 11:59PM.

Once your full submission is complete, your paper will undergo a series of checks in preparation for assessment.

Kind regards,

Richard Turner, PhD

rturner@plos.org

---

## [Decision Letter · Decision Letter 1]

25 Jun 2021

Dear Dr. Stover,

Thank you very much for submitting your manuscript "The Epidemiological Impact of the UNAIDS 2025 Targets for HIV" (PMEDICINE-D-21-02046R1) for consideration at PLOS Medicine. 

Your paper was discussed with an academic editor with relevant expertise, and sent to independent reviewers, including a statistical reviewer. The reviews are appended at the bottom of this email and any accompanying reviewer attachments can be seen via the link below:

[LINK]

In light of these reviews, we will not be able to accept the manuscript for publication in the journal in its current form, but we would like to invite you to submit a revised version that addresses the reviewers' and editors' comments fully. You will appreciate that we cannot make a decision about publication until we have seen the revised manuscript and your response, and we expect to seek re-review by one or more of the reviewers. 

We hope to receive your revised manuscript by Jul 15 2021 11:59PM. Please email us (plosmedicine@plos.org) if you have any questions or concerns.

Please let me know if you have any questions, and we look forward to receiving your revised manuscript. 

Sincerely,

Richard Turner, PhD

rturner@plos.org

Please add a study descriptor to the title following a colon, e.g., "...: A modeling study".

You may wish to amend the title to contain the phrase "HIV control" or "AIDS elimination". 

Please adapt the abstract to the three-part PLOS Medicine style. 

The final sentence of the "Methods and findings" subsection of the abstract should begin "Study limitations include ..." or similar and should quote 2-3 of the study's main limitations. 

After the abstract, we will need to ask you to add a new and accessible "Author summary" section in non-identical prose. You may find it helpful to consult one or two recent research papers published in PLOS Medicine to get a sense of the preferred style. 

A full point seems to be missing at line 133.

Please state, early in your Methods section, whether the study had a protocol or prespecified analysis plan, and if so attach the relevant document(s) as a supplementary file(s), referred to in the text. 

Please identify a suitable checklist for your study design, which might be TRIPOD, and if available include a completed checklist as a supplementary file. 

Please convert the "slides" into supplementary figures. 

To reference 1, for example, please add "UNAIDS" or another author, and specify an accessed date. 

Please update reference 16.

It may be that reference 24 will need to be removed, unless it is available as a preprint, say.

Please use the journal name abbreviation "PLoS ONE". 

Comments from the reviewers:

*** Reviewer #1: 

This article was a delight to read for a number of reasons. First, it is extensive in its breadth of subject and how up to date it is. Second, it is extremely timely as many of us in the HIV community are wondering about the longer term effects of COVID on HIV program success. Thirdly, this effort and the group of writers reflect the necessary collaboration to make meaningful estimates. The UNAIDS estimates have been a work in progress for many years and they have importantly evolved with increasingly better quality inputs and as a result, increasingly believable.

I have several comments for the authors to consider, all of which are to increase transparency and should not be considered mandatory.

1) Many countries do not have incidence estimates either ever or that are up to date. How did you manage estimating changes in infection rates in settings with poor knowledge of incidence?

2) Since condom use among clients of sex workers is typically extremely high in most settings, there is reason to believe that sex workers are less of a contributing factor to HIV among males that other risky behaviours. To what extent have you modelled sexual networks and how is defined?

3) You are, no doubt, aware of the convincing evidence that men have different (generally worse) outcomes than females for access to testing, treatment, and outcomes. Morna Cornell has well summarized this. Does your model adjust for this? It historically has not.

4) Previous UNAIDS estimates have relied heavily on South Africa estimates and applied these to other countries in Africa. To what extent is borrowing occurring in the analysis (not necessarily South Africa)?

*** Reviewer #2: 

This is a well-crafted paper continuing the authors' outstanding work on informing the global HIV response by pointing out the potential impact of its ambitious, sometimes aspirational targets. The authors took great care in adding two central new aspects: the impact of (not) improving societal enablers, inasmuch as this can be estimated; and the impact of COVID-19 related interruptions. The paper fulfils its purpose fully, but this stands next to a number of shortcomings and gaps in methods and method description mentioned below.

Major comments:

1. My main point of criticism is that it is computationally complex but conceptionally simple to tally the impact of maximising intervention coverage towards targets that were in turn set using the same models and their estimated intervention effect sizes. That these estimates of impact have proven useful for advocacy and fundraising does not diminish the fact that, by the same token, the impact itself remains unachieved, and, one could say, unachievable. While there is a role for this paper despite this central shortcoming, one would expect the authors to spend some time describing not only the size of the underachievement under past targets, but also some of the reasons, and, more importantly, caution against the possibility of the new set of targets not being achieved (or achievable) either. This would be especially welcome given that this year marks 20 years since the first such global impact estimation- 20 years of impacts that did not come to pass, or not by the timeline envisaged.

2. One of the reasons for past underachievement is underfunding. This paper clearly has a role even without a cost envelope component, but again would be more honest if affordability could be mentioned as an issue that might well stand between the targets and the estimated impact.

3. Introduction: Summarise progress against 75% reduction targets for main parameters by 2019.

4. Methods: The Methods section and Abstract does not specify that results were compared to Optima and AEM results for only seven countries; this is only mentioned in the Results section. Please specify this in the other sections as well. And given the size of its contribution to new infections and deaths and given that no Optima version exist for it, why were projections not also compared to a South African model, eg. Thembisa? How good is the fit of the South African GOALS model to Thembisa?

1. Given that there are explicit UNAIDS programme targets for transgender people, it would be good to have a justification for why they are not incorporated in GOALS (one would assume, just yet).

2. Line 243: "We assume that general population incidence for all other countries is below 0.3%" does not follow from there only being Naomi district-level incidence estimates for 25 countries. I would have expected how district-level differences in incidence were estimated off (available) national incidence estimates for the remaining countries, rather than a reversion to a single national-level default incidence value.

5. The team is to be commended for their careful approach to the estimation of the potential impact societal enablers. However, one wonders if the literature review in reference 22 would not allow the differentiation of one of the central assumptions, the % of PLHIV with internalised stigma, by region or even country.

6. What was the time frame (ie. which exact months and years- not just the duration) that COVID-19 related disruptions were assumed to take place?

Minor comments:

3. Abstract: Mention full time period modelled; add the 2025 targets; mention % change for infections and deaths if societal enablers were not achieved; mention what the short-term is (for the impact of COVID-19 related disruptions).

4. Line 133: A full stop missing after "[10-12]".

5. Line 197: "The program targets in the new plan" warrants a repetition of whose plan you are talking about.

6. The two statements in line 261 onwards, "Criminalization of sex work, same sex intercourse and drug use; stigma and discrimination and gender-based violence can result in low use of prevention, testing and treatment services. Interventions to address these issues can lead to more utilization of services", need some references.

7. Line 268: The "additional" seems misplaced here.

8. Line 284: Replace "might be" by "were estimated to be", and add the individual references after each estimate (even though they are also summarised in Table 5), since they seem to be directly informed by 1-2 studies each. 

9. Line 320: "women experiencing GBV" is not a great term. Consider rephrasing.

10. Line 344/ 348: Please add if these total reductions would be compared to a 2010 or 2019 baseline (also in the Abstract).

11. Line 356: "because of the additional non-AIDS-related deaths to PLHIV"- perhaps rather ""because of the increase in non-AIDS-related deaths in PLHIV"? Similarly, in line 372 change "to" to "in".

12. Line 365: Change "continuing" to "continue".

13. Page 23: You seem to not report the impact on deaths if societal enablers were not achieved, even though this is mentioned in the Abstract. Please add to the main text as well.

14. Perhaps add the finding of declining numbers of PLHIV on ART from 2023 into the Conclusion (and, if there is space, into the Abstract), as this is quite a central (and promising) finding as well.

15. Line 452: Perhaps replace "or" by "unless" (and adjust the rest of the sentence).

*** Reviewer #3: 

This study reports on efforts to use mathematical models to estimate the impact of achieving UNAIDS program targets for 2025, with the goal to eliminate AIDS as a public health threat by 2030. 

Comments:

"The main results presented here were produced using one or other of two different versions of the Goals model: the age-structured version of the model, Goals ASM, was applied to 39 countries from sub-Saharan Africa while the risk-structured version, Goals RSM, was used for 38 countries from other regions."

Can the authors please clarify why different approaches were applied here?

"Results from the Goals models were validated by comparing with outputs for select countries from two other models, Optima HIV, and the AIDS Epidemic Model (AEM)."

Can the authors also please expand on this text, clarifying why only select countries were compared? 

"The ranges used for the epidemiological parameters and the fitted values by country are provided in Supplementary Tables 1 and 2." 

The model parameters listed in Supplementary Table 2 (Fitted parameter values by country) appear to be fixed constants. Did the authors consider accounting for variability and uncertainty in there parameter estimates within the models? 

"The model is available for download free of charge from the Avenir Health web site as a module in the Spectrum software at https://avenirhealth.org/software-spectrum.php."

The authors have provided a useful and impactful resource, demonstrating good practice.

"This section describes how we estimated the impact of these three societal enablers. This approach was informed by a literature review that identified interventions with evidence of impact on each of the societal enablers and/or impact on HIV behaviors or outcomes [19]. "

Did the authors consider running any sensitivity analyses on the estimated value of these societal enablers? 

Of note, in Slide 13 there is no data plotted for the scenario of 'no progress' within the AIDS related deaths?

In general, can the authors please quantify and communicate the uncertainty surrounding point estimates attained from each model throughout the results section, as well as in all figures?

Please can the authors label the y axis in all figures?

Overall, this is a well written manuscript with a clear and concise description of each type of model.

The mathematical formulation would be welcome as supplementary material.

*** Reviewer #4: 

This is a useful high-level modeling analysis that led to target setting to achieve the updated UNAIDS 95/95/95 goals. 

I still have several concerns. Specifically, I am missing the following:

-Commentary on the feasibility of each of the different goals/targets (aside from the 'softer' targets, which were discussed to some degree)

-Given lines 51-54: A critical self-assessment of the value in modeling in goal setting: how did modeling fit into the process of target setting? What are the benefits and drawbacks of this approach?

-Modeling has been used in the past in UNAIDS target settings: what was learned from previous processes and updated consequently? What has modeling gotten wrong previously? How has modeling to improve guidelines improved? 

-A critical assessment and discussion of the drawbacks of the Goals model - just because the model is in line with results from Optima and AEM doesn't make the results inherently correct.

-Presumably, the people remaining unreached within each of the targets/goals may differ from those who have already been reached. Are those reached with an intervention assumed to be the same (in terms of risk of onward infection) to those who are not reached within each of the targets? The remaining populations not targeted by different interventions may possibly disproportionately contribute to onward infections. If this is the case, how are the results affected?

-Sensitivity analyses: based on the uncertainties surrounding the inputs and expected impact of various interventions, I would expect a robust sensitivity analysis

***

[LINK]

---

## [Decision Letter · Decision Letter 2]

21 Sep 2021

Dear Dr. Stover,

Thank you very much for re-submitting your manuscript "Modeling the Epidemiological Impact of the UNAIDS 2025 Targets on Ending AIDS as a Public Health Threat by 2030" (PMEDICINE-D-21-02046R2) for consideration at PLOS Medicine.

I have discussed the paper with editorial colleagues and our academic editor, and it was also seen again by four reviewers. I am pleased to tell you that, provided the remaining editorial and production issues are fully dealt with, we expect to be able to accept the paper for publication in the journal.

[LINK]

Please let me know if you have any questions, and we look forward to receiving the revised manuscript.   

Sincerely,

Richard Turner, PhD

rturner@plos.org

Requests from Editors:

The title needs a study descriptor, and we suggest modifying the title to: "Estimating the Epidemiological Impact of the UNAIDS 2025 Targets to End AIDS as a Public Health Threat by 2030: A Modeling Study".

In the abstract, an additional sentence or two are needed early in the "Methods and Findings" subsection to explain the methodology, including what the "Goals model" is.

At line 91, please adapt the text to note that the present paper is part of a PLOS Medicine Collection around the UNAIDS targets.

At line 499, please make the section heading "Discussion".

Noting references 14 & 24, please list 6 author names in reference citations, followed by "et al.".

Noting reference 29, please ensure that all citations have full access details. 

Thank you for including the GATHER checklist: please rename this "S1_GATHER_Checklist" or similar and refer to it as such early in the Methods section (main text). 

We suggest providing a list of supplementary files at the end of the paper. 

Are we right in thinking that the files labeled "slide 12" and "slide 13" correspond to the supplementary figures? Please rename the files as appropriate if so. 

Please ensure that all figures have labeled axes. 

Comments from academic editor: 

The authors should add a response to reviewer 2's comment 3.

Comments from Reviewers:

*** Reviewer #1: 

The authors have adequately addressed all points I consider to be important. 

*** Reviewer #2: 

Thanks for the careful revisions to your paper. Almost all my comments have been sufficiently taken into account, with the exception of the following:

- Comments 1, 2 and 3: It is good to know that other papers cover the aspects mentioned, but very few people will read the entire special issue. Please add a short summary of each of these aspects with a reference to the other papers where needed.

- Comment 6: Please add the details on how many countries you had Naomi estimates for into the paper.

- Comment 14: Please still add the individual references after each estimate in Line 284 and following.

- Comment 15: I did not want to be prescriptive, but the point here was to move away from an active term describing women subjected to GBV, given that they most often don't have much choice in the matter. If I may make a suggestion, however, perhaps replace the term in Line 320 by exactly that, "women subjected to GBV".

*** Reviewer #3: 

The authors have satisfactorily responded to each comment in turn, and have strengthened the reporting of the study by more clearly presenting and discussing uncertainty. 

*** Reviewer #4: 

The authors have addressed all of my concerns.

***

[LINK]

---

## [Editor Report · Decision Letter 3]

1 Oct 2021

Dear Dr Stover, 

On behalf of my colleagues and the Academic Editor, Dr Rosen, I am pleased to inform you that we have agreed to publish your manuscript "Modelling the Epidemiological Impact of the UNAIDS 2025 Targets on Ending AIDS as a Public Health Threat by 2030" (PMEDICINE-D-21-02046R3) in PLOS Medicine.

Prior to final acceptance, please: remove reference 19, unless this has been accepted for publication; and provide labels for the graph axes, which we cannot see in the PDF.

PRESS

Sincerely, 

Richard Turner, PhD 

rturner@plos.org